# Exploring the transmission of cognitive task information through optimal brain pathways

**Zhengdong Wang**[1], **Yifeixue Yang**[1], **Ziyi Huang**[1], **Wanyun Zhao**[1], **Kaiqiang Su**[1], **Hengcheng Zhu**[2], **Dazhi Yin**[1,3]*

1 Shanghai Key Laboratory of Brain Functional Genomics (Ministry of Education), Affiliated Mental Health Center (ECNU), School of Psychology and Cognitive Science, East China Normal University, Shanghai, China, 2 Division of Biostatistics, University of Minnesota, Minneapolis, Minnesota, United States of America, 3 Shanghai Changning Mental Health Center, Shanghai, China

* dzyin@psy.ecnu.edu.cn

## Abstract

Understanding the large-scale information processing that underlies complex human cognition is the central goal of cognitive neuroscience. While emerging activity flow models demonstrate that cognitive task information is transferred by interregional functional or structural connectivity, graph-theory-based models typically assume that neural communication occurs via the shortest path of brain networks. However, whether the shortest path is the optimal route for empirical cognitive information transmission remains unclear. Based on a large-scale activity flow mapping framework, we found that the performance of activity flow prediction with the shortest path was significantly lower than that with the direct path. The shortest path routing was superior to other network communication strategies, including search information, path ensembles, and navigation. Intriguingly, the shortest path outperformed the direct path in activity flow prediction when the physical distance constraint and asymmetric routing contribution were simultaneously considered. This study not only challenges the shortest path assumption through empirical network models but also suggests that cognitive task information routing is constrained by the spatial and functional embedding of the brain network.

## Author summary

A fundamental concern of cognitive neuroscience is the emergence of the complex brain functions in humans. The transmission of neural signals in the brain is thought to be fundamental to cognition. However, it remains unclear how does cognitive information transmit effectively from the perspective of large-scale brain networks. While graph theory is innately dedicated to characterizing brain networks, there is still a gap between graph routing protocols and cognitive task activity. To this end, we test whether the graph-theory-based shortest path outperforms direct path and decentralized network communication routes leveraging empirical activity flow modeling. Results demonstrate that shortest path routing is superior to other network communication strategies in activity flow prediction, but inferior to direct path routing. Importantly, the incorporation of spatial distance and functional asymmetry improves prediction accuracy. This study

**Data availability statement:** Data availability HCP dataset used in this study is publicly available in the repository of Human Connectome Project (https://db.humanconnectome.org/data/projects/HCP_1200) (Requiring a simple sign-on to access and registering for an account is free.). For our dataset, the fMRI metadata for validation analysis are available in https://osf.io/cm5f2. Code availability The main codes used for this study are available in https://osf.io/cm5f2. The functions for search information and navigation are available in the Brain Connectivity Toolbox (https://www.nitrc.org/projects/bct).

**Funding:** This work was supported by the National Natural Science Foundation of China (32271096 to D.Z.Y.), the Science and Technology Innovation 2030-Major Projects (2021ZD0200500 to D.Z.Y.), and the Fundamental Research Funds for the Central Universities (D.Z.Y.). The funders had no role in study design, data collection and analysis, decision to publish, or preparation of the manuscript.

**Competing interests:** The authors have declared that no competing interests exist.

not only sheds light on the mechanistic relationships between cognitive task activation, resting-state network topology, spatial geometry, and functional embedding, but also advances our understanding of complex communication mechanisms of the human brain.

## Introduction

There is a growing consensus that cognitive functions originate from intricate interactions among neural elements in the brain [1–7]. These interactions may occur in the form of neural communication at different spatiotemporal scales, such as synaptic transmission, interactions between neuronal populations, and interregional communication. However, the exact brain communication mechanism remains elusive, especially for large-scale information processing that underlies human flexibility in performing various cognitive tasks. Additionally, cognitive neuroscience studies usually consider two types of basic functional brain activity (i.e., task-evoked and spontaneous/intrinsic) separately [8–10], which is not conducive to understanding information processing within a unified framework.

Inspired by artificial neural network models, a seminal study proposed a large-scale activity flow model [11], which assumed that the task-induced activity of a given region could be predicted by the linear summation of the activations of all other regions weighted by their resting-state functional connectivity (FC) with the target region. While FC has been calculated using different methods, such as standard Pearson's correlation, multiple regression, and probabilistic correlation, cognitive task information is hypothesized to be transmitted through direct FC between regions [12–20]. Considering that resting-state network connectivity can be predicted by the structural connectome [21–23], activity flow mapping has also been tested using structural connections as routes [24]. However, a recent study using computational models has shown that the prediction of task-evoked brain activity is mainly contributed by resting-state FC rather than structural connections [25], even though anatomical connectivity was demonstrated to predict brain activity in a functional specialized region [26, 27]. Moreover, although physically linked structural connections can facilitate direct signal transmission, FC may exist between distant and anatomically unconnected regions and reflect polysynaptic information communication [28–32]. Nevertheless, whether cognitive task information is transmitted via multistep or indirect paths remains largely unknown.

Based on the graph theoretical framework [33], the brain is often modeled as functional or structural networks [34–38]. Many graph measures, such as characteristic path length, global efficiency, and betweenness centrality, assume that interregional communication is routed exclusively via the shortest path of brain networks [39, 40]. Computational modeling work has also shown that the shortest path structure of the human connectome accelerates the spread of information cascades [41]. However, the shortest path routing has been controversial for the theoretical models of brain function because it requires a biologically implausible assumption that neural elements have knowledge of global network topology [42, 43]. Accordingly, several network communication models that do not require centralized knowledge, such as diffusion models [44–47], path ensembles [48], and navigation [49], have been proposed to characterize neural signaling. Although these network communication strategies usually enable better inference of structure-function correspondences, little is known about whether they are superior to the shortest path routing for diverse cognitive information transmission.

Importantly, the primate brains are organized hierarchically in the temporal and spatial domains [50, 51]. However, the vast majority of existing studies have characterized path lengths based on undirected or symmetric network connections owing to the limitations of noninvasive measuring techniques. Thus, there is the potential assumption that a connection

makes the same contribution to the transmission of information at its two endpoints. Considering that the functional role of a region is largely determined by its connectional fingerprint [52–54], it is necessary to consider the functional embedding of a connection, particularly if its two endpoints belong to different cortical hierarchies. Through network communication models, Seguin et al. recently revealed send-receive communication asymmetry across the cortical hierarchy based on an undirected structural connectome [55]. Specifically, the unimodal areas exhibited greater send-receive asymmetry than did multimodal areas. Moreover, emerging evidence suggests that spatial geometry influences the topology of the human connectome, thereby conferring functional advantages [56–62]. Nevertheless, there is a lack of empirical evidence of the influence of functional and spatial embedding on cognitive information transmission.

While previous activity flow modeling studies only used interregional FC as a path, in this study, we aimed to test whether cognitive task information is transmitted via multistep routes, such as the shortest path (Fig 1). We further compared distinct decentralized network communication strategies, including search information, path ensembles, and navigation, with centralized shortest path routing in the context of cognitive information transmission. Finally, we examined the effects of asymmetric routing contributions and physical distance constraints on cognitive information transmission by considering the functional and spatial embedding of a route. Our objective was to leverage the predictive performance of cognitive activity flow to make inferences regarding which routing strategies govern information transmission. We hypothesized that the shortest path routing may not outperform direct FC routes in predicting cognitive information transmission; however, regardless of the routing strategy, we speculated that the prediction of cognitive information transmission would be improved when functional and spatial embeddings are considered.

## Results

The imaging datasets used for the main analysis were obtained from the publicly available Human Connectome Project (HCP). Specifically, we adopted a main dataset that included 100 unrelated participants with all functional magnetic resonance imaging (fMRI) data (i.e., four resting-state fMRI runs). One hundred unrelated participants were included in the replication dataset. In addition, we also adopted a "reduced dataset," comprising the same participants

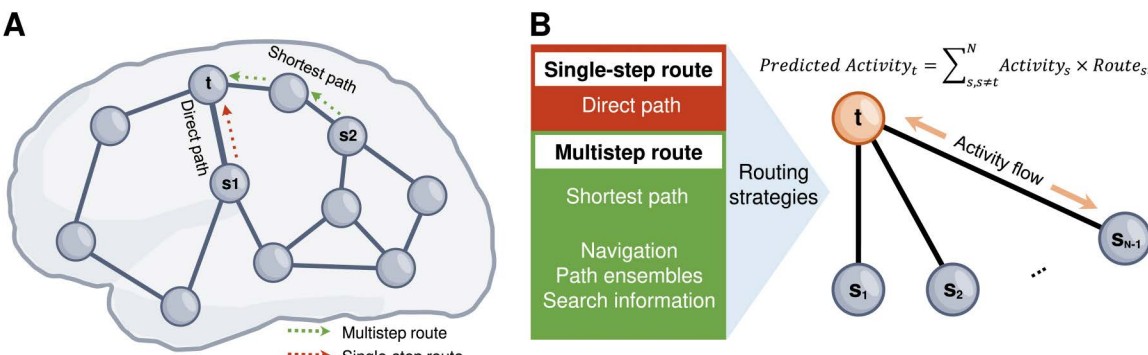

**Fig 1. Schematic diagram of two types of typical neural communication routes (i.e., single-step and multistep). (A)** Illustration of the direct path (single-step) from the source node *s1* to the target node *t* (red dash line) and the shortest path (multistep) from nodes *s2* to *t* (green dash line). The brain render was drawn by hand using Microsoft Visio. **(B)** Illustration of activity flow modeling with different routing strategies. *Route_st* indicates direct functional connectivity, shortest path, or other network communication strategies (i.e., navigation, path ensembles, and search information) between nodes *s* and *t*.

included in the main dataset but with only one resting-state fMRI run. Based on the graph theoretical framework, previous studies often treated the human brain as a sparse network (weighted or binary) [34,63] with a range of density thresholds (e.g., 2%-50%). Here, all types of activity flow routes were derived from sparse functional brain networks. For the commonly used density threshold of 15%, an exemplar procedure of activity flow modeling based on the direct path (i.e., FC) and shortest path (i.e., $SPL_{wei}$, shortest path length based on a weighted network; $SPL_{bin}$, shortest path length based on a binary network) was used (Fig 2). Notably, the $SPL_{wei}$ and $SPL_{bin}$ matrices were denser than the original FC matrix, which is attributed to the indirect paths considered. Additionally, the following statistical analyses were performed on the area under the curve (AUC) of the prediction accuracy ($r$) across different density thresholds.

## Direct path outperforms shortest path for cognitive information transmission

Although activity flow modeling often adopts a direct path (FC), we first tested whether the graph-theory-based shortest path yielded a better cognitive activity flow prediction. For main dataset, we found that the mean prediction accuracies ranged from $r = 0.48$ to $r = 0.53$ for the direct path, from $r = 0.44$ to $r = 0.53$ for the $SPL_{wei}$, and from $r = 0.33$ to $r = 0.47$ for the $SPL_{bin}$ over all density thresholds. For replication dataset, we found that the mean prediction accuracies ranged from $r = 0.49$ to $r = 0.56$ for the direct path, from $r = 0.46$ to $r = 0.56$ for the $SPL_{wei}$, and from $r = 0.36$ to $r = 0.50$ for the $SPL_{bin}$ over all density thresholds. For reduced dataset, we found that the mean prediction accuracies ranged from $r = 0.45$ to $r = 0.50$ for the direct path, from $r = 0.41$ to $r = 0.49$ for the $SPL_{wei}$, and from $r = 0.28$ to $r = 0.43$ for the $SPL_{bin}$ over all density thresholds.

Through one-way repeated measures analysis of variance (ANOVA), we found significant differences in AUC values of the prediction accuracy based on three different activity flow routes (main dataset: $F_{2, 196} = 916.26$, $\eta^2 = 0.90$, $p < 0.001$; replication dataset: $F_{2, 198} = 1113.71$, $\eta^2 = 0.92$, $p < 0.001$; and reduced dataset: $F_{2, 196} = 1224.41$, $\eta^2 = 0.93$, $p < 0.001$). Post-hoc analyses revealed that the AUC values of the prediction accuracy based on the shortest path ($SPL_{wei}$ and $SPL_{bin}$) were significantly lower than those based on the direct path for all datasets (all $ps < 0.05$, Bonferroni corrected). Moreover, the AUC values of the prediction accuracy based on $SPL_{bin}$ were significantly lower than those based on $SPL_{wei}$ for all datasets (all $ps < 0.05$, Bonferroni corrected) (Fig 3). Notably, the active flow prediction based on the direct path and $SPL_{wei}$ became comparable when network density beyond 25%. This is consistent with extending the density threshold to 100% (S1 Fig). These findings indicate that the direct path is superior to the shortest path in terms of cognitive information transmission, especially for low network density. Additionally, $SPL_{bin}$ routing, which showed the worst performance, suggests that the link weights of brain networks are important for cognitive information transmission, whereas brain networks are often binarized for graph theory analysis.

## Shortest path length routing superior to stepwise shortest path protocol in cognitive information transmission

We also simulated activity flows over multiple steps to test the efficacy of multi-step activity flow processes. Briefly, we performed stepwise calculation for the transformation of cognitive task activation from source to target according to the shortest path. For main dataset, we found that the mean prediction accuracies ranged from $r = 0.44$ to $r = 0.53$ for the $SPL_{wei}$ and from $r = 0.30$ to $r = 0.52$ for the stepwise $SP_{wei}$ over all density thresholds. For replication dataset, we found that the mean prediction accuracies ranged from $r = 0.46$ to $r = 0.56$ for the $SPL_{wei}$ and from $r = 0.30$ to $r = 0.54$ for the stepwise $SP_{wei}$ over all density thresholds. For reduced dataset, we found that the mean prediction accuracies ranged from $r = 0.41$ to $r =$

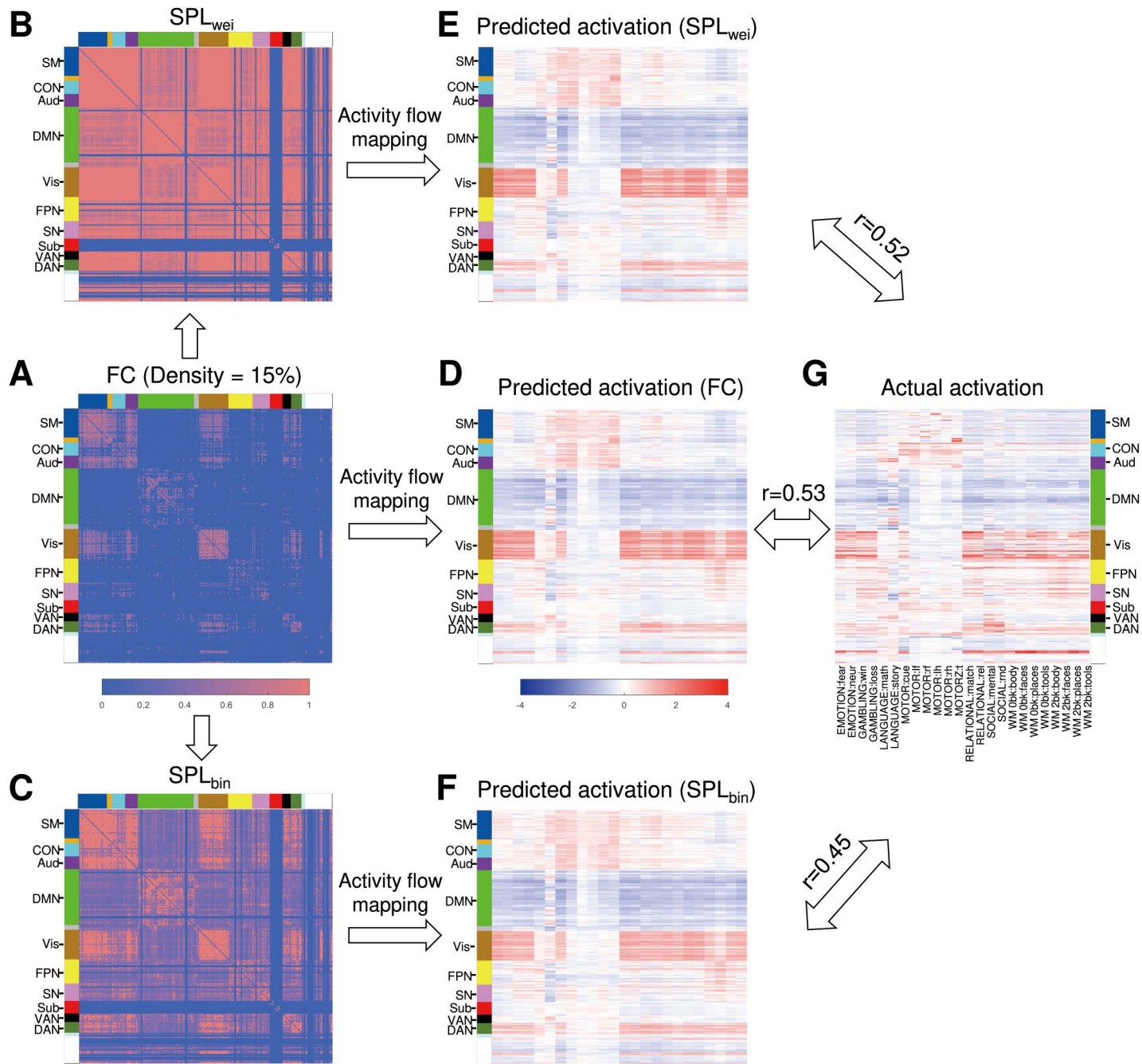

**Fig 2. An exemplar procedure for activity flow modeling based on the direct and shortest paths at a network density of 15%. (A)** Mean FC (Pearson's correlation) matrix across participants. **(B-C)** Mean shortest path length matrices were derived from the weighted and binary FC matrices using Dijkstra's algorithm. **(D-F)** The predicted activations of 24 task contrasts by activity flow modeling with the routes of FC, $SPL_{wei}$, and $SPL_{bin}$, respectively. **(G)** The actual activations of 24 task contrasts. $r$ values denote mean prediction accuracy across participants. FC, functional connectivity; $SPL_{wei}$, shortest path length based on weighted network; $SPL_{bin}$, shortest path length based on binary network; Aud, auditory network; CON, cingulo-opercular network; DAN, dorsal attention network; DMN, default mode network; FPN, frontoparietal network; SM, sensorimotor network; SN, salience network; Sub, subcortical network; VAN, ventral attention network; Vis, visual network.

0.49 for the $SPL_{wei}$, and from $r = 0.19$ to $r = 0.46$ for the stepwise $SP_{wei}$ over all density thresholds. Through paired sample $t$-tests, we found that the AUC values of the prediction accuracy with stepwise activity flow processes along the shortest path was significantly lower than that

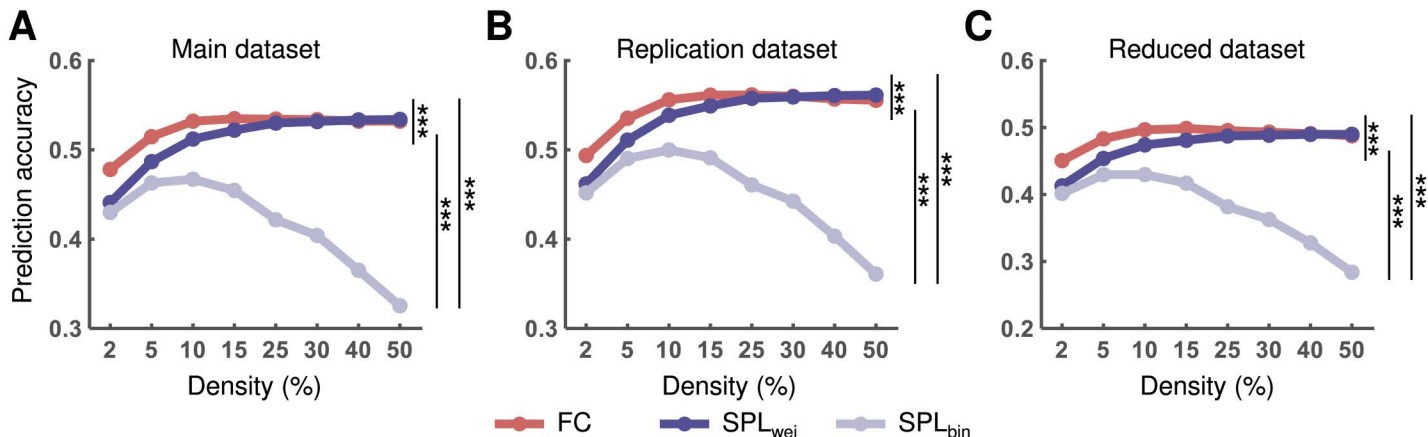

**Fig 3. Accuracy of the activity flow prediction based on the direct (FC) and shortest (SPL$_{wei}$ and SPL$_{bin}$) paths.** Statistical comparisons were performed for the main (**A**), replication (**B**), and reduced (**C**) datasets. Statistical significance was identified based on the area under the curve (AUC) across all density thresholds. FC, functional connectivity; SPL$_{wei}$, shortest path length based on weighted network; SPL$_{bin}$, shortest path length based on binary network. ***$p < 0.001$ ($p < 0.05$, Bonferroni corrected).

with the shortest path length directly for all datasets (all *ps* < 0.001, Bonferroni corrected) (Fig 4). This result suggests that the shortest path length routing is superior to the stepwise shortest path protocol in cognitive information transmission.

## Shortest path routing outperforms other network communication strategies for cognitive information transmission

We subsequently tested whether the network communication strategies (i.e., search information, path ensembles, and navigation) (Fig 5A-5C) were superior to shortest path routing for cognitive information transmission. For the weighted network, repeated measures ANOVA showed significant differences in AUC values of the prediction accuracy based on different network communication strategies (main dataset: $F_{3, 294} = 1251.09$, $\eta^2 = 0.93$, $p < 0.001$; replication dataset: $F_{3, 297} = 1435.45$, $\eta^2 = 0.94$, $p < 0.001$; and reduced dataset: $F_{3, 294} = 667.86$, $\eta^2 = 0.87$, $p < 0.001$). Post-hoc analyses revealed that the AUC values of prediction accuracy based on the shortest path were significantly (all *ps* < 0.05, Bonferroni corrected) greater than those based on the network communication strategies for all datasets, except for navigation routing in reduced dataset (Fig 5D-5F). A consistent result was observed for the binary network, except that there was no significant difference in prediction accuracy between shortest path routing and navigation routing for the main and reduced datasets (Fig 5G-5I). Although there was a significant difference, we found that navigation routing was comparable to the shortest path routing for activity flow prediction. These findings suggest that the shortest path routing outperforms other network communication strategies for cognitive information transmission.

## Effects of spatial embedding of routes on cognitive information transmission

To test whether physical distance constrains the functional path between two regions, thereby influencing cognitive information transmission, we performed activity flow modeling based on routes that were modulated by Euclidean distance (Fig 6A). For all datasets, we found that the accuracy of activity flow prediction was significantly enhanced (all *ps* < 0.05, Bonferroni-corrected) for both the direct and shortest paths when the physical distance was considered

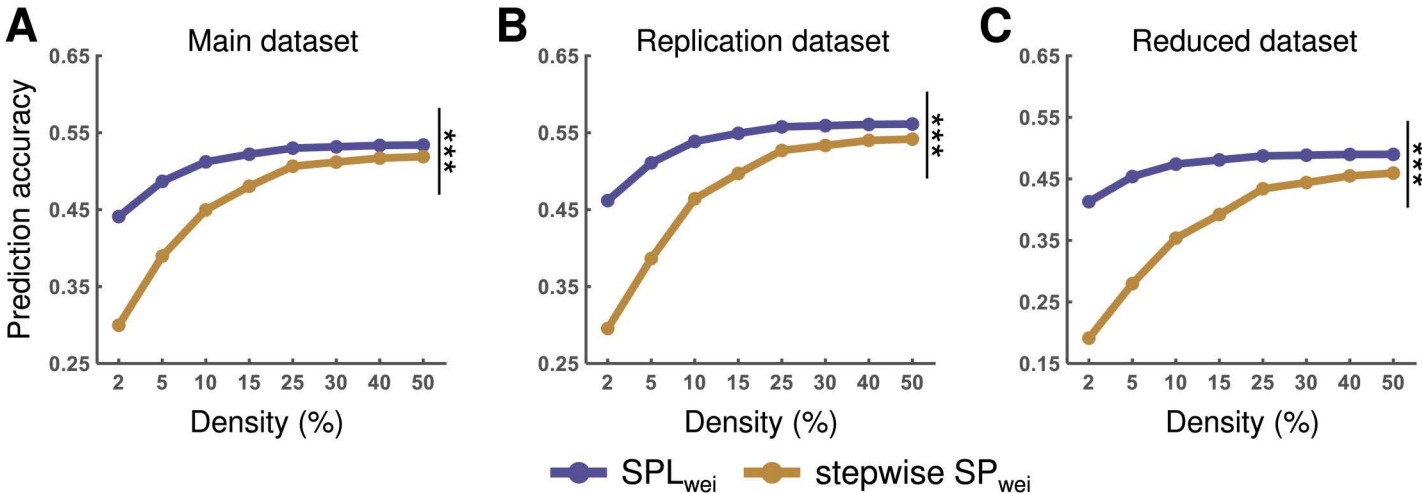

**Fig 4. Accuracy of the activity flow prediction based on the shortest (SPL$_{wei}$) and stepwise shortest (SP$_{wei}$) paths.** Statistical comparisons were performed for the main **(A)**, replication **(B)**, and reduced **(C)** datasets. Statistical significance was identified based on the area under the curve (AUC) across all density thresholds. SPL$_{wei}$, shortest path length based on weighted network; stepwise SP$_{wei}$, shortest path identified by weighted network and simulating stepwise activity flow processes. ***$p < 0.001$ ($p < 0.05$, Bonferroni corrected).

(Fig 6B-6D). This result indicates that the spatial embedding of routes may play a crucial role in cognitive information transmission.

### Effects of functional embedding of routes on cognitive information transmission

Considering that a connection likely contributes asymmetrically to routing to its two end-points (Fig 6E), we rescaled the common connection by dividing its weight by the mean of all connection weights of a certain endpoint. Thus, an asymmetric routing matrix was obtained. For all datasets, we found that the accuracy of activity flow prediction was significantly enhanced (all $ps < 0.05$, Bonferroni corrected) for both the direct and shortest paths when the asymmetric routing contribution was considered (Fig 6F-6H). This indicates that the functional embedding of routes may also play a crucial role in cognitive information transmission. Additionally, the shortest path (SPL$_{wei}$) was superior to the direct path for activity flow prediction after considering the functional embedding of routes.

### Greater performance gains in activity flow prediction after combining spatial and functional embedding of routes

On considering that the spatial or functional embedding of routes improves activity flow prediction, we further tested whether greater performance gains could be achieved by considering them simultaneously. Repeated measures ANOVA showed significant differences in performance of activity flow prediction when spatial embedding, functional embedding, or the both were considered for the main (direct path: $F_{2, 196} = 389.68$, $\eta^2 = 0.80$, $p < 0.001$; SPL$_{wei}$: $F_{2, 196} = 257.39$, $\eta^2 = 0.72$, $p < 0.001$; SPL$_{bin}$: $F_{2, 196} = 1470.37$, $\eta^2 = 0.94$, $p < 0.001$), replication (direct path: $F_{2, 198} = 480.61$, $\eta^2 = 0.83$, $p < 0.001$; SPL$_{wei}$: $F_{2, 198} = 233.02$, $\eta^2 = 0.70$, $p < 0.001$; SPL$_{bin}$: $F_{2, 198} = 1609.16$, $\eta^2 = 0.94$, $p < 0.001$), and reduced (direct path: $F_{2, 196} = 616.37$, $\eta^2 = 0.86$, $p < 0.001$; SPL$_{wei}$: $F_{2, 196} = 398.33$, $\eta^2 = 0.80$, $p < 0.001$; SPL$_{bin}$: $F_{2, 196} = 1463.67$, $\eta^2 = 0.94$, $p < 0.001$) datasets. Post-hoc analyses further indicated greater performance gains in activity flow prediction of the shortest path on considering both spatial and functional embedding than on considering either

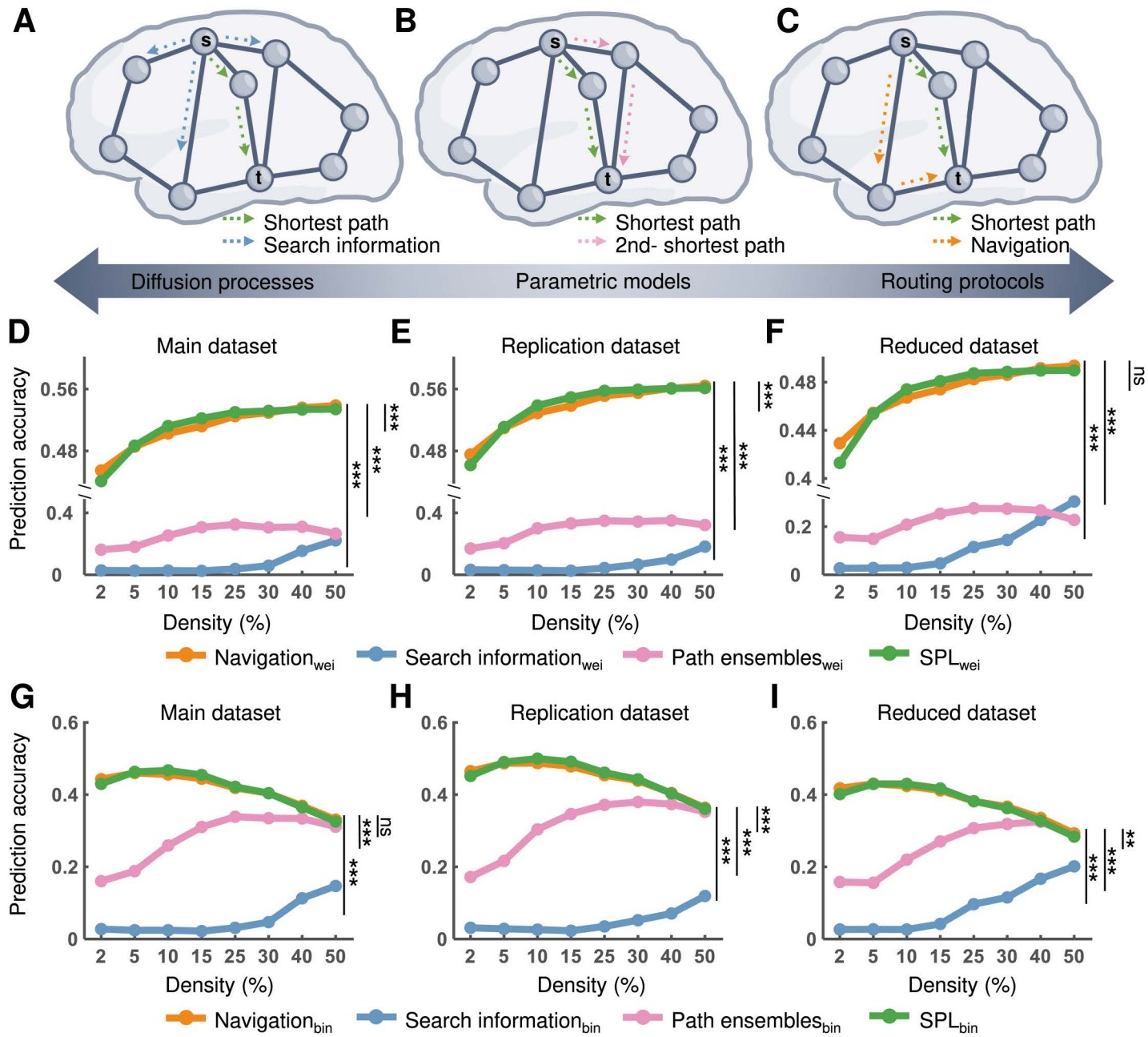

**Fig 5. Accuracy of the activity flow prediction based on the shortest path routing and other network communication strategies. (A-C)** Illustration of search information, path ensembles, and navigation, respectively. Search information considers the divergence of information via the shortest path and relates to the probability of efficient path accessibility. Path ensembles integrate multiple efficient paths (*k*-SPL) in consideration of the trade-off between transmission cost and delay (here, *k* = 2 was adopted). Navigation path denotes moving to the neighboring node that is closest to the target node guided by physical distance. The brain render was drawn by hand using Microsoft Visio. **(D-F)** For the weighted network, the accuracy of activity flow prediction based on the shortest path routing and other network communication strategies for all datasets. **(G-I)** For the binary network, the accuracy of activity flow prediction based on the shortest path routing and other network communication strategies for all datasets. Statistical significance was identified based on the area under the curve (AUC) across all denisty thresholds. $X_{wei}$, routing metric X calculated based on the weighted network; $X_{bin}$, routing metric calculated based on the binary network; ns, nonsignificant. ***$p < 0.001$ ($p < 0.05$, Bonferroni corrected).

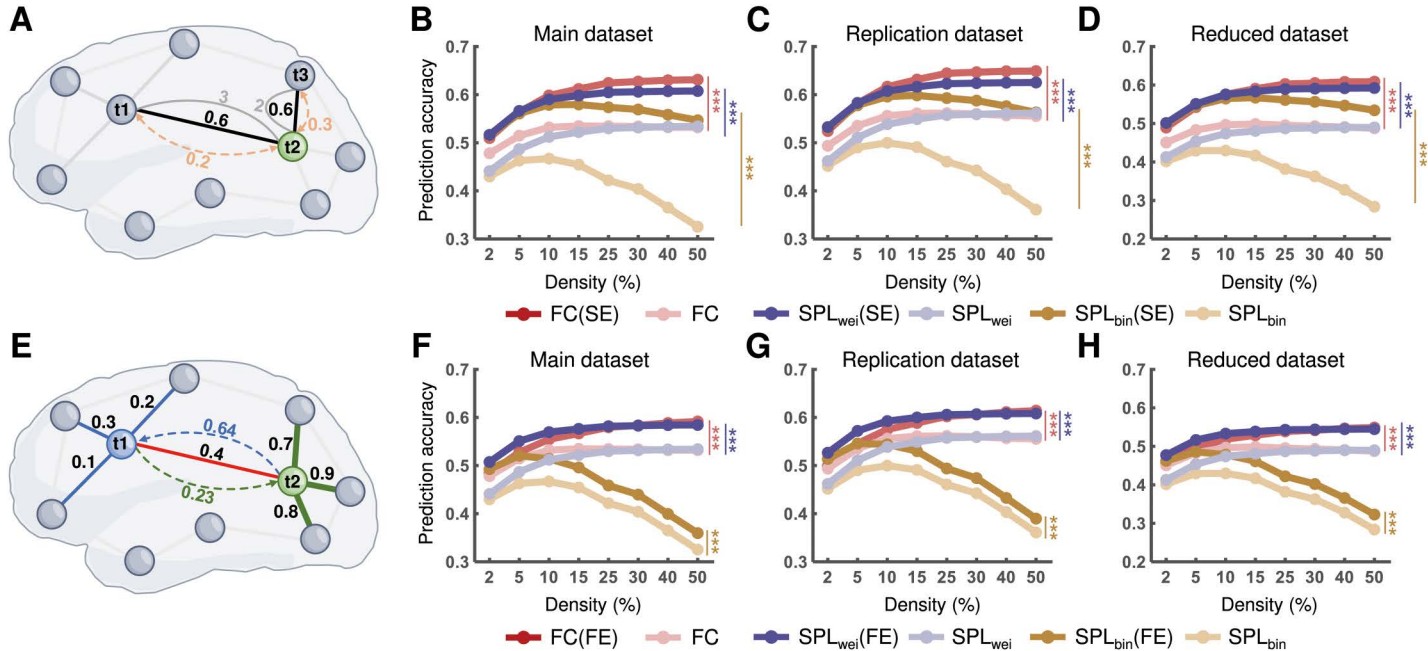

**Fig 6. Effects of spatial and functional embedding of routes on activity flow prediction. (A)** Schematic diagram of the influence of physical distance on routes. When two functional connections ($FC_{12}$ and $FC_{23}$) have the same weight (= 0.6) but different physical distance (3 and 2, respectively), the modulated weights of $FC_{12}$ and $FC_{23}$ would be 0.2 and 0.3, respectively. **(B-D)** Statistical comparisons of the accuracy of activity flow prediction before and after considering the spatial embedding of routes for all datasets. **(E)** Schematic diagram of the influence of functional embedding on routes. There is a common connection between nodes *t1* and *t2* with the weight of 0.4. If weights of the other connections of node *t1* (e.g., 0.1, 0.2, and 0.3) are lower than those of node *t2* (e.g., 0.7, 0.8, and 0.9), their common connection likely contributes asymmetrically to routing to nodes *t1* and *t2*. On dividing the weights by the mean of all connection weights of a certain endpoint, the rescaled weights would be 0.64 and 0.23 for nodes *t1* and *t2*, respectively. **(F-H)** Statistical comparisons of the accuracy of activity flow prediction before and after considering functional embedding of routes for all datasets. Statistical significance was identified based on the area under the curve (AUC) across all density thresholds. The brain render was drawn by hand using Microsoft Visio. FC, functional connectivity; SE, spatial embedding; FE, functional embedding; $SPL_{wei}$, shortest path length based on weighted network; $SPL_{bin}$, shortest path length based on binary network. ***$p < 0.001$ ($p < 0.05$, Bonferroni corrected).

one for all datasets (all $ps < 0.05$, Bonferroni corrected). In contrast, the direct path showed comparable performance gains in activity flow prediction when considering spatial and functional embedding simultaneously and when considering only spatial embedding (Fig 7). These findings indicate that the shortest path routing may yield greater performance gains in activity flow prediction when considering spatial and functional embedding simultaneously.

On comparing the performance in activity flow prediction among the routes that combined spatial and functional embedding, for main dataset, we found that the mean prediction accuracies ranged from $r = 0.51$ to $r = 0.64$ for the direct path, from $r = 0.53$ to $r = 0.63$ for the $SPL_{wei}$, and from $r = 0.53$ to $r = 0.60$ for the $SPL_{bin}$ over all density thresholds. For replication dataset, we found that the mean prediction accuracies ranged from $r = 0.52$ to $r = 0.66$ for the direct path, from $r = 0.55$ to $r = 0.65$ for the $SPL_{wei}$, and from $r = 0.55$ to $r = 0.62$ for the $SPL_{bin}$ over all density thresholds. For reduced dataset, we found that the mean prediction accuracies ranged from $r = 0.49$ to $r = 0.62$ for the direct path, from $r = 0.51$ to $r = 0.61$ for the $SPL_{wei}$, and from $r = 0.51$ to $r = 0.69$ for the $SPL_{bin}$ over all density thresholds.

Furthermore, repeated measures ANOVA showed significant differences in the AUC values of prediction accuracy based on different routes (main dataset: $F_{2, 196} = 169.97$, $\eta^2 = 0.63$, $p < 0.001$; replication dataset: $F_{2, 198} = 229.93$, $\eta^2 = 0.70$, $p < 0.001$; and reduced dataset: $F_{2, 196} = 132.47$, $\eta^2 = 0.58$, $p < 0.001$). Post-hoc analyses further revealed that the accuracy of activity flow prediction based on $SPL_{wei}$ was greater than that based on $SPL_{bin}$ and the direct path for all datasets (all $ps < 0.05$, Bonferroni corrected). The accuracy of activity flow prediction based on $SPL_{bin}$ was

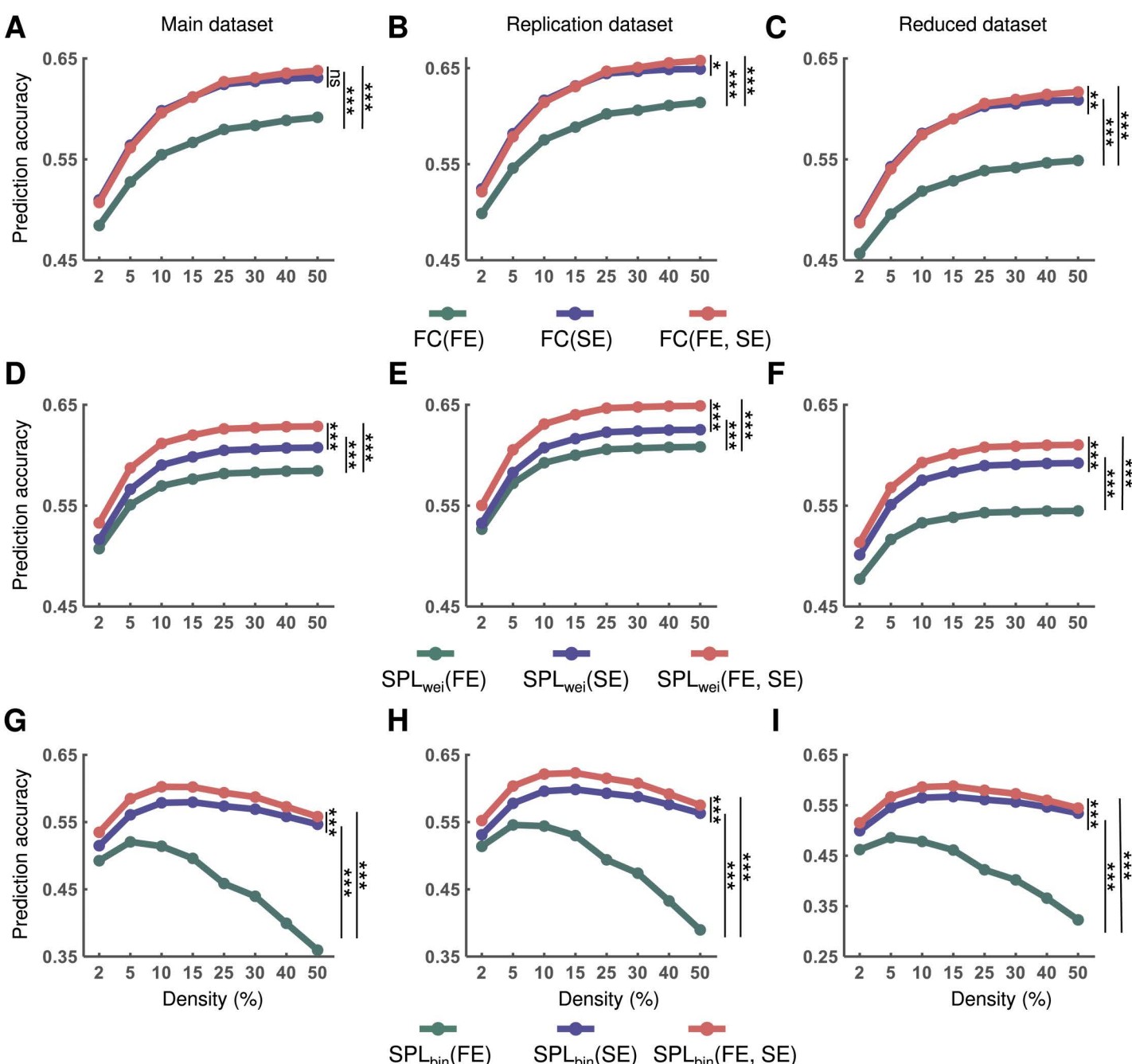

**Fig 7. Performance gains in activity flow prediction after combining spatial and functional embedding of routes.** Accuracy of activity flow prediction based on FC (**A**), FC(SE) (**B**), and FC(FE, SE) (**C**) for different datasets. Accuracy of activity flow prediction based on SPL (**D**), $SPL_{wei}$(SE) (**E**), and $SPL_{wei}$(FE, SE) (**F**) for different datasets. Accuracy of activity flow prediction based on SPL (**G**), $SPL_{bin}$(SE) (**H**), and $SPL_{bin}$(FE, SE) (**I**) for different datasets. Statistical significance was identified based on the area under the curve (AUC) across all density thresholds. FC, functional connectivity; FE, functional embedding; SE, spatial embedding; $SPL_{wei}$, shortest path length based on weighted network; $SPL_{bin}$, shortest path length based on binary network; ns, nonsignificant. *$p < 0.05$, ***$p < 0.001$ ($p < 0.05$, Bonferroni corrected).

greater than that based on the direct path only for the reduced dataset ($p < 0.05$, Bonferroni corrected) (Fig 8). This result suggests that the shortest path routing might be optimal for cognitive information transmission after modulation by spatial and functional embedding.

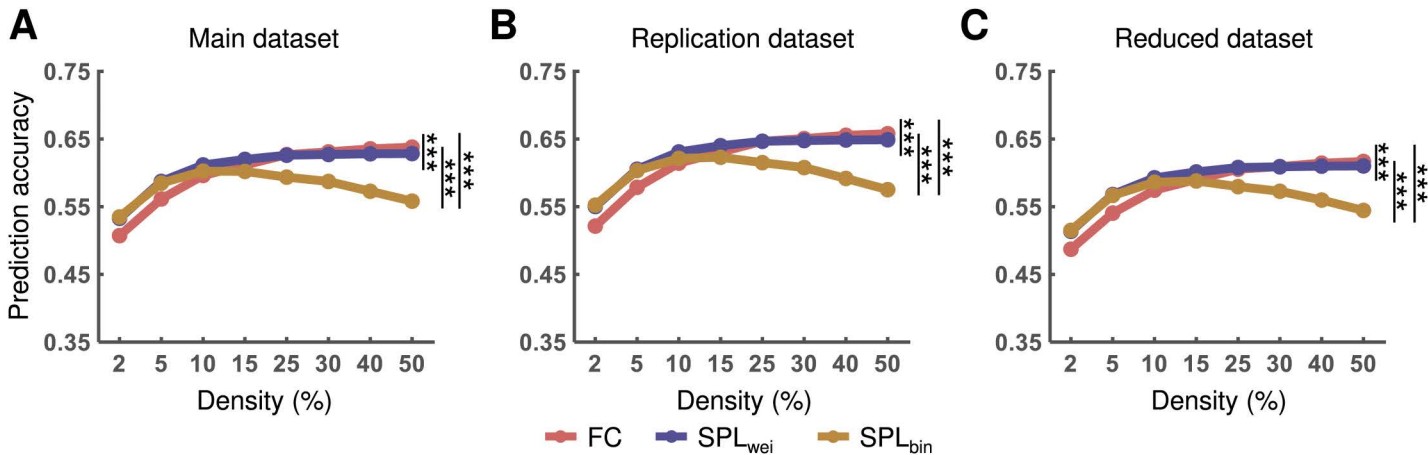

**Fig 8. Performance of the activity flow prediction with different routes that combined spatial and functional embedding.** The prediction accuracy is shown for the main (**A**), replication (**B**), and reduced (**C**) datasets. Statistical significance was identified based on the area under curve (AUC) across all density thresholds. FC, functional connectivity; FE, functional embedding; SE, spatial embedding; $SPL_{wei}$, shortest path length based on weighted network; $SPL_{bin}$, shortest path length based on binary network; ns, nonsignificant. $*p < 0.05$, $***p < 0.001$ ($p < 0.05$, Bonferroni corrected).

## Validation of our main findings

To validate our main findings, we performed the main analyses on an independent dataset created by our group [19]. Specifically, we recruited 51 participants who performed the *n*-back task with different working memory loads (i.e., 2-back, 3-back, and 4-back). Consistent with the HCP dataset, repeated measures ANOVA showed significant differences in the AUC values of prediction accuracy based on different routes ($F_{2, 100} = 362.30$, $\eta^2 = 0.88$, $p < 0.001$). Post-hoc analyses consistently revealed that the AUC values of prediction accuracy based on the direct path were significantly higher than those based on the shortest path ($SPL_{wei}$ and $SPL_{bin}$) (all $ps < 0.05$, Bonferroni corrected) (S2 Fig).

For different network communication models, significant differences were also observed in the AUC values of prediction accuracy using repeated measures ANOVA (weighted network: $F_{3, 150} = 141.37$, $\eta^2 = 0.74$, $p < 0.001$; binary network: $F_{3, 150} = 83.78$, $\eta^2 = 0.63$, $p < 0.001$). Furthermore, post-hoc analyses showed that the accuracy based on search information and path ensembles was consistently lower than that based on the shortest path for both weighted and binary networks (all $ps < 0.05$, Bonferroni corrected), except for the search information with weighted networks (S3 Fig). However, the AUC values of prediction accuracy based on navigation were significantly higher than those based on the shortest path for both the weighted and binary networks. Notably, the difference mainly occurred in sparse networks, which is consistent with the findings for the HCP dataset.

On considering the spatial or functional embedding of routes, the accuracy of activity flow prediction improved significantly (all $ps < 0.05$, Bonferroni corrected) (S4 Fig). Moreover, repeated measures ANOVA consistently showed that performance gains in activity flow prediction with spatial and functional embedding were significantly different for our dataset (direct path: $F_{2, 100} = 65.83$, $\eta^2 = 0.57$, $p < 0.001$; $SPL_{wei}$: $F_{2, 100} = 57.64$, $\eta^2 = 0.54$, $p < 0.001$; $SPL_{bin}$: $F_{2, 100} = 188.66$, $\eta^2 = 0.79$, $p < 0.001$). Post-hoc analyses revealed greater performance gains in activity flow prediction for both the shortest path and the direct path when spatial and functional embedding were considered simultaneously (all $ps < 0.05$, Bonferroni corrected) (S5 Fig). Moreover, repeated measures ANOVA showed significant differences in the AUC values of activity flow prediction among different routes when spatial and functional embedding

were considered simultaneously ($F_{2, 100} = 14.73$, $\eta^2 = 0.23$, $p < 0.001$). In the post-hoc analyses, we consistently found that the AUC values of activity flow prediction based on $SPL_{wei}$ were significantly higher than those based on the direct path (all $ps < 0.05$, Bonferroni corrected) (S6 Fig). These results indicate that our main findings can be generalized to other datasets.

Additionally, the parameter $k = 2$ was selected for path ensembles [48] in the main analysis. To test the influence of this parameter, we performed activity flow prediction based on path ensembles with $k = 10$. Consistently, the performance of activity flow prediction based on path ensembles was significantly lower than that based on the shortest path (all $ps < 0.05$, Bonferroni-corrected) for both the HCP dataset (S7 Fig) and our dataset (S8 Fig). This result suggests that our main findings are robust to the parameter $k$.

To keep consistency with the original FC network, we also conducted an analysis by applying the same density thresholds to the SPL networks. Here, we considered three situations as follows: (1) a sparsification of SPL networks according to path weights directly. We found that the prediction accuracy with the shortest path was still lower than that with the direct path, and this difference mainly occurred in the case of high network density. (2) a sparsification of SPL networks with the direct and indirect paths preserved proportionally. The prediction accuracy with the SPL networks was dramatically lower than that with the FC network across all density thresholds. (3) a sparsification of SPL networks with only indirect paths preserved. We found that the prediction accuracy with the SPL networks was dramatically lower than that with the FC network across all density thresholds (S9 Fig). These findings suggest that the overall performance of activity flow prediction is determined by routing protocols rather than network density.

## Discussion

Understanding how the human brain performs complex cognitive functions is a fundamental challenge in the field of cognitive neuroscience. Based on the graph theory, the brain has been demonstrated to be an economic, small-world, modular network [64–66] that enables the balancing of functional segregation and integration to facilitate flexible cognition [67, 68]. However, these organizing principles are often extrapolated from the network topology, generally assuming that neural signals are transmitted through the shortest path and that the brain is a centralized system. Beyond topological properties, emerging network communication models have questioned whether neural signaling occurs via the shortest path of a structural connectome, and they underscore the view that the brain is a decentralized system [5,42]. In the present study, to test whether cognitive information is transmitted via the shortest path, we examined multiple routing strategies (both decentralized and centralized) for activity flow prediction, which can provide information on a mechanistic relationship between task-evoked activation and intrinsic network topology [11,19]. Moreover, we explored the effects of spatial and functional embedding of routes on activity flow prediction. This study not only offers empirical network models of cognitive information flow for testing the shortest path assumption but also links brain function to network topology and spatial geometry.

Specifically, we found that activity flow prediction with a direct FC path was significantly better than that with the shortest path. This suggests that cognitive task information may not be transmitted via the shortest path in the resting-state brain network. Although intuitively appealing, the shortest path assumption that underlies typical graph measures appears problematic [37]. Regarding the biological plausibility of polysynaptic neural signaling, a previous study indicated that intrinsic FC can reflect the polysynaptic anatomical pathway [32]. Thus, it helps to confirm that direct FC is likely sufficient for routing cognitive task information. Previous studies on brain network communication [42] have suggested that information

propagation throughout the brain requires only local knowledge. Our results further challenge the shortest path assumption by providing empirical evidence of routing cognitive information.

However, we found that the shortest path routing was superior to other network communication strategies (i.e., search information, path ensembles, and navigation). Using a model of cascade spreading, prior study also showed that the shortest paths in the network play a key architectural feature in neural communication [41]. In contrast, many studies have revealed that network communication models enable better prediction of resting-state FC from the structural connectome [45,46,55,69–71] than does the shortest path strategy. Moreover, a recent study demonstrated that communication dynamics can explain more variance in the propagation of focal direct electrical stimulation than transmission via the shortest path [30]. It seems that decentralized communication strategies offer more plausible neural signaling routes, such as anatomically unconnected paths, which may lead to an improvement in structure-function correspondence. These controversial findings about centralized and decentralized communication strategies are at least partly due to testing different questions. Nevertheless, cognitive information transmission between regions is likely governed by a parsimonious routing protocol (direct or short path), which also ensures the fidelity of neural communication. Therefore, network communication strategies may not be optimal for routing cognitive information when performing specific tasks.

Notably, navigation routing showed better predictive performance for cognitive activity flow at very sparse connection densities (e.g., density = 2%) compared to that of the shortest path routing, while the overall predictive performance was slightly lower. Similarly, Seguin et al. reported that the navigation efficiency of brain networks is comparable to that of the shortest path routing (> 80% of optimal efficiency) [49]. Moreover, navigation routing relies on both the network topology and spatial geometry, which may be conducive to efficient decentralized communication. A previous study also indicated that the probability of a functional connection can be sufficiently modeled by a spatial distance penalty and topological term that favors links between regions sharing nearest neighbors [60]. Moreover, functional organization, such as the connectivity gradient, is tightly linked to the spatial geometry of the brain network [72]. Therefore, we speculate that the superiority of navigation routing for a sparse network is attributable to the incorporation of a spatial constraint. We consistently found that activity flow prediction significantly improved for both the direct and shortest paths when a spatial distance penalty was considered. The results suggest that spatial embedding plays an important role in the transmission of cognitive information.

Additionally, we observed that activity flow prediction significantly improved for both the direct and shortest paths when the asymmetric routing contribution was considered. Compared with using standard FC (i.e., Pearson's correlation) with a symmetric routing contribution, previous studies have consistently shown that using a asymmetric routing contribution inferred from multiple regression FC [11], probabilistic correlation [15], and directed FC [20] methods can improve the performance of activity flow models. Instead, the asymmetric routing proposed in this study considers the functional embedding of a connection, which complies with the theory that the functional role of a region is largely determined by its connectional profile [52–54]. Interestingly, we noticed that asymmetric routing for the shortest paths can improve the activity flow prediction more significantly. On combining spatial and functional embedding for the shortest paths, we observed greater performance gains in activity flow prediction compared with when either one was considered. Therefore, our findings not only emphasize the importance of the functional embedding of routes in information transmission but also imply that the functional role of a region can be manifested by a multistep or polysynaptic connection profile.

On considering spatial and functional embedding simultaneously, the shortest paths (especially for sparse networks) outperformed the direct paths in activity flow prediction. This indicates that the shortest path routing for cognitive task information should be modulated by spatial and functional embedding. Therefore, the inference of neural signaling based only on network topology may be insufficient when following the traditional graph theoretical framework. Nonetheless, path-length-based graph measures have been successfully related to individual differences in cognitive performance [73–75] and disease symptoms [76–78]. Although the shortest paths allow information to be efficiently routed throughout the brain, such a putative protocol may be impractical and constrained by the geometric and functional architecture [37,49]. In the future, it will be necessary to combine network topology, spatial geometry, and functional embedding to infer information propagation in the brain.

In analogy with artificial neural networks, recent studies have modeled activity flows over multiple steps [79, 80]. However, these models mainly focus on specific nervous systems (e.g., visual processing system), which simulate decentralized information processing. In contrast, the shortest path routing requires the knowledge of global network topology. While progressive information propagation seems biologically plausible, it may be not the case for the shortest path routing at the level of whole brain. On the other hand, many graph measures based solely on the shortest path length can well characterize brain hubs that play a critical role in information communication underlying diverse cognitive functions [81]. Therefore, it may be appropriate to use the shortest path length as a weight for routing cognitive information transmission.

Regarding the effects of network density, we found that the direct path was superior to the shortest path, mainly in the case of low network density. This result indicates that indirect paths may play adverse effects on prediction accuracy, when the original FC network is sparse. As the network density increases, many indirect paths may be replaced by direct paths, making the prediction accuracy comparable. Through a sparsification of SPL networks for keeping the same density as the original FC network, we found that the prediction accuracy with the shortest path was also lower than that with the direct path, and this difference mainly occurred in the case of high network density. It is possible that the removed SPL edges are mainly indirect paths for low network density but direct paths for high network density, because many indirect paths may be replaced by direct paths as the network density increases. These findings suggest that performance in activity flow prediction with different routing protocols are not dominated by network density. Moreover, for graph theory analysis, graph measures and differences between groups (e.g., patients versus controls) may vary with network density [63,82]. Consistently, we found that the direct and shortest path results converge at the high density threshold. Theoretically, as the network density increases, many indirect paths may be replaced by direct paths, making the prediction accuracy comparable.

This study had some limitations. First, previous studies have mainly evaluated routing efficiency based on structural connectome predictions of functional network organization [45,46,69]. In contrast, this study leveraged activity flow modeling to characterize the routing efficiency. Although these large-scale neural models have been extensively tested using empirical data, the limited spatiotemporal resolution of noninvasive neuroimaging may prevent adequate inference of neural signaling. Second, while Pearson correlation is a commonly used method for measuring FC, it may produce spurious connections and cannot characterize causal interactions [83], which could affect the inference of activity flows. A recent study used the Bayesian network-based Peter-Clark algorithm to infer directed and causal activity flow routes [20]; alternative directed FC approaches can also be applied, such as dynamic causal models [84, 85], Granger causality [86], and transfer entropy [87]. However, to date, there are no methods that can make perfect causal inferences using noninvasive

neuroimaging techniques, and the causal connectivity inferred by different methods can be inconsistent [88–90]. Third, although a previous study has demonstrated improvements in activity flow prediction when using task-state FC [91], resting-state FC is most commonly used for activity flow routes. This is because the resting-state FC configuration is independent of the current task of interest. Thus, the model can be generalized across different task states [79]. Moreover, inferring the FC using data from another state may overcome model overfitting to some extent [92]. Finally, several FC methods have been proposed to improve activity flow prediction [15,20]. However, in this study, we used the standard Pearson's correlation approach, which requires few assumptions and is most frequently employed for the construction of functional brain networks. Thus, it enables better linking of the activity flow mapping framework and graph theoretical framework.

In conclusion, our findings revealed that the direct path outperformed the shortest path in cognitive information transmission. However, the shortest path routing was superior to other network communication strategies, including search information, path ensembles, and navigation. Interestingly, the shortest path outperformed the direct path in activity flow prediction when the physical distance constraint and asymmetric routing contribution were simultaneously considered. This study not only challenges the shortest path assumption through the use of empirical network models but also suggests that routed information transmission might be modulated by spatial and functional embedding. The current study sheds light on the mechanistic relationships between cognitive task activation, resting-state network topology, spatial geometry, and functional embedding, which could aid in making inferences regarding which routing strategies govern information transmission for the individuals with neuropsychiatric disorders.

## Materials and methods

### Ethics statement

This study was approved by the Ethics Committee of East China Normal University (Approval Number: HR1-0148-2023). Written informed consent was obtained from all participants of HCP dataset and our dataset.

### MRI datasets

The MRI data used for our main analysis were collected from the HCP (https://www.human-connectome.org). Following previous studies [11,15], this study also included a main dataset (100 unrelated participants) and replication dataset (100 unrelated participants) with all fMRI sessions (four resting-state fMRI runs). To examine the effect of the number of data points, a reduced dataset with the same participants as those in the main dataset but with only one resting-state fMRI run was adopted. One participant was excluded from the main dataset due to poor image quality. The average age of the participants was 29 years (range, 22–36 years) for the main dataset, and 46% were male. For the replication dataset, the average age was 28 years (range, 22–35 years), and 50% were male. For the HCP data, the institutional review board of each site approved data collection, and informed consent was obtained from each participant. Additionally, to validate our main findings, we used another independent dataset comprising 51 participants (31 females; average age, 21.8 years; range, 19–27 years) who performed the *n*-back task with different working memory loads [19].

### Image acquisition and preprocessing

For the HCP dataset, all fMRI scans were acquired using a modified 3 T Siemens Skyra MRI scanner and adopted a multiband echo-planar imaging sequence with a 32-channel head coil

[93]. The specific parameters of the pulse sequence were as follows: repetition time (TR) = 720 ms, echo time (TE) = 33.1 ms, acceleration factor = 8, flip angle (FA) = 52°, bandwidth = 2290 Hz/Px, in-plane field of view (FOV) = 208 × 180 mm², 72 slices, and 2.0 mm isotropic voxels. Data were collected over 2 days. Resting-state fMRI scans (28 min, two runs) were performed with the eyes open and fixed, followed by task fMRI scans performed (30 min, one run for each task) daily. There were seven tasks: emotion cognition, gambling reward, language, motor, relational reasoning, social cognition, and working memory. Further details on the data collection process have been reported in previous studies [94, 95].

We used minimal preprocessing of the fMRI data, including standard procedure implementation (standard normalization and template, motion correction, and intensity normalization) [96]. The SPM12 toolbox (http://www.fil.ion.ucl.ac.uk/spm) and REST software [97] were used to preprocess the resting-state and task-state fMRI data. For the resting-state fMRI data, we discarded the first 10 volumes of each run for signal equilibrium to help the participants adapt to the experimental environment. Several nuisance time series, including motion estimates and cerebrospinal fluid and white matter signals, were removed using linear regression. Global signal regression was not conducted due to the ongoing debate [98, 99] on this preprocessing step. Linear trend removal, bandpass filtering (0.01–0.08 Hz), and spatial smoothing (full width at half maximum of 4 mm) were performed. For task-state fMRI data, we removed motion estimates using linear regression and then conducted spatial smoothing using a 4-mm Gaussian filter.

For our dataset, both resting-state and task-state fMRI data were obtained using a 3.0 Tesla MRI scanner at the Shanghai Key Laboratory of Magnetic Resonance, East China Normal University. The main scanning parameters were as follows: TR = 2530 ms, TE = 2.98 ms, FA = 7°, and FOV = 256 × 256 mm² for structural images and TR = 2000 ms, TE = 30 ms, FA = 90°, and FOV = 224 × 224 mm² for functional images. More details regarding the image acquisition and preprocessing methods used can be found in our recent study [19].

## Construction of resting-state brain networks

To maintain consistency with previous studies [11,15], a parcellation scheme comprising 264 functional regions of interest (ROIs) [100] was used. For each ROI, the mean time series was obtained by averaging all voxels within the ROI. The Pearson's correlation for the time series of the paired ROIs was then evaluated (i.e., FC). Subsequently, Fisher's z-transformation was performed to normalize the correlation values. Finally, a functional brain network represented as a symmetric matrix, $F \in R^{N \times N}$, was obtained for each participant. Element $F_{ij}$ in matrix $F$ represents the connection weight between nodes $i$ and $j$.

The graph-theory-based brain network analysis often adopts a thresholding strategy to remove spurious/weak connections and assumes that the brain is composed of sparse networks. Since there is currently no definitive criterion for selecting a single threshold, a range of density thresholds (e.g., 2%-50%) is commonly used [40,63,100–102]. Therefore, for our main analyses, we used a range of density thresholds from 2% to 50%, with extending to 100% for a validation. In addition, negative connections may have a meaning in theory (inhibitory connectivity). However, it remains to be demonstrated whether the negative connections estimated by fMRI represent inhibitory connectivity. Following previous studies [15,39], all negative connections were set to 0.

## Activity flow modeling

Based on the pioneering study on activity flow mapping [11], cognitive task activation of a given brain region can be predicted by linearly summing the activation of all other ROIs, weighted by resting-state FC. The formulation of this model is expressed by Eq. 1:

$$E_j = \sum_{i \neq j \in V} A_i F_{ij} \qquad (1)$$

where $E_j$ is the predicted activation for the target region $j$ in a given task, $A_i$ denotes the actual activation (i.e., beta-weights or difference in beta-weights between two conditions) of region $i$ in a given task, which is estimated by general linear model. And, $F_{ij}$ is the weight of the FC between regions $i$ and $j$. The activity flow model is implemented through leave-one-region-out simulations for the comparison between prediction and actual activity.

The accuracy of activity flow prediction was evaluated using a similarity measure ($r$). We used Pearson's correlation to calculate the similarity between the predicted and actual activation across all brain regions individually; subsequently, the correlations were averaged across all participants. Notably, traditional activity flow modeling adopts a direct path (FC) as a route. To test whether cognitive information was transmitted via the shortest path or other decentralized network communication strategies, we performed activity flow modeling using different routing protocols (i.e., replacing $F_{ij}$ with other path estimates).

### Shortest path length calculation

To identify the shortest paths, we first defined the topological distance between two nodes of the brain network. A commonly used transformation from network connectivity to topological distance is $T_{uv} = 1/F_{uv}$, where $F_{uv}$ represents the connectivity weight between nodes $u$ and $v$, and $T_{uv}$ denotes the topological distance. In other words, the stronger the connection between nodes, the shorter the topological distance. Moreover, $T_{uv} = \infty$ indicates that nodes $u$ and $v$ are not connected. In practice, we set it to zero if there is no connection. Dijkstra's algorithm was implemented to compute the SPL based on a topological distance matrix. It is worth noting that we did not exclude the true shortest path (the direct path) when considering the shortest path. In other words, the "shortest path" communication protocol may include both the direct path and multi-step shortest path.

Notably, the above mentioned SPL calculation was based on the weighted network; therefore, we named it $SPL_{wei}$. For the binary network, Dijkstra's algorithm was directly implemented to compute the SPL, which was termed $SPL_{bin}$. Additionally, we used $L_{ij}$ to denote the shortest path length ($SPL_{wei}/SPL_{bin}$) between regions $i$ and $j$. Therefore, activity flow prediction based on the SPL was formulated as follows:

$$E_j = \sum_{i \neq j \in V} A_i \frac{1}{L_{ij}} \qquad (2)$$

In addition, we have conducted an analysis that keeps the density of the resulting networks consistent by applying the same density thresholds to the SPL networks. Here, we considered three situations as follows: (1) a sparsification of SPL networks according to path weights directly; (2) a sparsification of SPL networks with direct and indirect paths preserved proportionally; and (3) a sparsification of SPL networks with only indirect paths preserved.

### Simulation of multi-step activity flow processes for the shortest path routing

In our main analyses, we simply added a connection from a source to a target with a weight reflecting the shortest path length between them. Considering the biological plausibility, we also investigated whether the activity flows propagate in a progressive manner. Inspired by recent studies [79, 80], we simulated activity flows over multiple steps to test the efficacy of multi-step activity flow processes. Specifically, we performed stepwise calculation for the

transformation of cognitive task activation from source to target according to the shortest path. As a consequence, activity flow prediction based on the stepwise shortest path simulation was formulated as follows:

$$E_t = \sum_{s \neq t \in V} A_s \prod_{u,v \in \Omega_{s \to t}} \frac{1}{T_{uv}} \tag{3}$$

Where $\Omega_{s \to t} = \{s, i, j, \ldots, k, t\}$ denotes the corresponding nodes sequence of the shortest path from source node $s$ to target node $t$. $E_t$ is the predicted activation for the node $t$. $A_s$ denotes actual activation of region $i$ for a given task. $T_{uv} = \{T_{si}, T_{ij}, \ldots, T_{kt}\}$ is the topological distance along the shortest path nodes sequence from $s$ to $t$.

## Network communication models

In addition to the centralized shortest path routing, four decentralized network communication strategies were tested as activity flow routes [42]. Specifically, the diffusion processes suggest that brain signals spread simultaneously along multiple paths, usually known as random walkers or agents. This signal broadcast does not require global knowledge of the entire connectome of an individual neural element [5]. Therefore, the diffusion processes often require multiple retransmissions from one node to the destination, which may require more time to search for an efficient path. Search information, as a typical diffusion process, is related to the probability that an agent walks randomly from the source node to the target node along the shortest path, quantifying the effort or amount of information required to access the path. If $B$ denotes the binary network connectivity matrix, the item $B_{ij} = 1$ if $F_{ij} > 0$, or $B_{ij} = 0$ otherwise. Given source node $s$ and target node $t$, sequenced edges of traveling the shortest path from $s$ to $t$ can be described as $\pi_{s \to t} = \{F_{si}, F_{ij}, \cdots F_{kt}\}$ or $\{B_{si}, B_{ij}, \cdots B_{kt}\}$. Furthermore, the corresponding sequence of nodes is $\Omega_{s \to t} = \{s, i, j, \cdots k, t\}$, with the number of traveling steps and nodes as $|\pi_{s \to t}| = V$ and $|\Omega_{s \to t}| = V + 1$, respectively. Hence, the probability of traveling along the shortest path from $s$ to $t$ is formulated as follows:

$$P(\pi_{s \to t}) = \prod_{i \in \Omega_{s \to t}^*} \frac{\pi_{i \to t}^{(1)}}{F_i} \tag{4}$$

where $\pi_{i \to t}^{(1)}$ is the first element of the path $\pi_{i \to t}$, and $\Omega_{s \to t}^*$ denotes the sequence of all nodes of the shortest path except the target node $s$ (i.e., $\Omega_{s \to t}^* = \{s, i, j, \cdots k\}$). Moreover, the information cost for accessing the path is $S(\pi_{s \to t}) = -log_2\left(P(\pi_{s \to t})\right)$. In addition, $F$ is replaced with $B$ for the binary network. A more detailed description can be found in previous studies [46,55].

Navigation is a routing protocol that employs the greedy strategy. It does not require global information on network topology but relies on local knowledge for individual nodes. Briefly, following simple rules, walking to the next node from the current node depends on which neighboring node is closest to the destination node, as measured by the physical distance (i.e., the Euclidean distance). This process does not guarantee a successful search for the optimal path but instead gets stuck in a loop when no neighbors are closer to the target node. However, navigational communication strategies continue to work efficiently in large-scale systems such as social, transportation, and biological networks [43,49,103]. The navigation protocol is motivated by the neural embedding of the brain in the physical space and strong relationship between the structural connection strength and Euclidean distance [57,62]. Therefore, navigation routing is often guided by the physical distance between two nodes, such as the Euclidean distance. In practice, a Euclidean distance matrix, $D \in R^{N \times N}$, is calculated for a standard parcellation template with three-dimensional coordinates. Subsequently,

following the greedy strategy guided by $D$, the walk starts from node $s$ to the next node $i$, which is closest to the target node $t$, until the target is reached through the node sequence $\{s,i,j,\cdots,k,t\}$. This process is repeated until any two nodes are traversed. Finally, the navigation path length matrix, $\Lambda \in R^{N \times N}$ is generated for each participant. It is worth noting that $\Lambda_{st}$ represents the summation of FC weight along the path identified by navigation. Specifically, $\Lambda_{st} = F_{si} + F_{ij} + \cdots + F_{kt}$. $\Lambda_{st} = \infty$ if node $s$ and node $t$ are disconnected utilizing the navigation strategy. The navigable path length is the number of steps between two nodes in the binary network. Functions for search information and navigation are available in the Brain Connectivity Toolbox (https://www.nitrc.org/projects/bct).

Parametric models, such as path ensembles, are hybrid strategies that consider not only information transmission delay but also the energy cost for searching for an efficient path. In contrast to the conventional shortest path routing, which involves a single path, the path ensemble model makes an assumption that information communication does not exactly occur along the shortest path but may involve a combination with other $k$ ($k \geq 2$) most efficient paths. Previous studies have demonstrated that alternative paths are selected for information communication in many complex networks [104,105]. Notably, parameter $k$ is the key to the path ensemble model, which still relies on the global knowledge of the network topology if $k$ is small. Otherwise, path construction/search with an extremely large $k$ will significantly increase the energy cost. Therefore, we adopted $k = 2$ for our main analysis, which means that the second shortest path was also considered. Furthermore, $k$ = 10 was examined for the validation analysis. Specifically, $k$-shortest paths should first be calculated using Yen's algorithm [106] for each pair of nodes in $B$ transformed from the FC matrix. Next, a three-dimensional matrix, $G \in R^{N \times N \times K}$, is obtained for each participant, where $N$ and $K$ denote the number of regions and shortest path in a network, respectively. Item $d\left(\pi_{s,t}^{k}\right)$ indicates the length of $k$-th shortest path between nodes $s$ and $t$ in $G$. Finally, path ensembles measure the constitution of $k$ efficient paths for a participant, as described in Eq. 5.

$$\mathbb{D}_{k}\left(s,t\right) = \sum \check{P}\left(\pi_{s,t}^{k}\right) d\left(\pi_{s,t}^{k}\right) \tag{5}$$

Here, $\check{P}\left(\pi_{s,t}^{k}\right)$ presents the probability of following the $k$-th shortest path from node $s$ to $t$ under a random walk, which can be formulated as follows:

$$P\left(\pi_{s,t}^{k}\right) = \prod_{e_{u,v} \in \pi_{s,t}^{k}} \frac{F_{u,v}}{\sum_{j} F_{u,j}} \tag{6}$$

where $F_{uv}$ denotes the functional connection weight between nodes $u$ and $v$. Eq. 6 indicates the product of each weight proportion along the $k$-th shortest path, and $\check{P}$ is normalized through $\check{P}\left(\pi_{s,t}^{k}\right) = P\left(\pi_{s,t}^{k}\right) / \sum_{K} P\left(\pi_{s,t}^{k}\right)$ ensuring that $\sum_{K}^{K=1} P\left(\pi_{s,t}^{k}\right) = 1$. That is, the $\check{P}\left(\pi_{s,t}^{k}\right)$ measures accessibility of a given path, $\pi_{s,t}^{k}$, in relation to other paths.

## Spatial embedding of activity flow routes

The network topology of the human brain has been demonstrated to be effectively characterized by a latent space rather than only a measure of the shortest path [107,108]. Furthermore, existing studies have shown that generative network models incorporating spatial constraints enable more accurate reproduction of the topological characteristics of brain networks [60,109–111]. Accordingly, spatial embedding may serve as a constraint in the regulation of functional signal transmission. In this study, we employed a simple method to embed physical information into activity flow modeling. Specifically, given the distance matrix, $D \in R^{N \times N}$

, calculated by spatial coordinates of network nodes, spatial embedding can be incorporated into the classical activity flow model as follows:

$$E_j = \sum_{i \neq j \in V} A_i F_{ij} \frac{1}{D_{ij}} \tag{7}$$

### Functional embedding of activity flow routes

Extensive evidence suggests a hierarchical organization of the brain [51]. A pathway between two regions may result in an asymmetric routing contribution to the two endpoints. In addition to the inference of directed functional interactions [20], a recent study revealed send-receive communication asymmetry across the cortical hierarchy based on an undirected structural connectome [55]. Considering that the functional role of a region is largely determined by its connection profile [52] in this study, we rescaled the connection weight by considering its functional embedding. Specifically, functional embedding was implemented by calculating the scale factor for any pair of nodes within a network and then the connection weight was reassigned. It was calculated using the following formula:

$$SC_{ij} = \frac{F_{ij}}{\left(\sum_t^N F_{it}\right) / C_i} \times F_{ij} \tag{8}$$

where $SC_{ij}$ is the rescaled connection weight for a given target node $i$, $C_i$ is the number of links to node $i$, and $F_{ij}$ represents FC weight. It is noteworthy that $SC_{ij}$ may differ from $SC_{ji}$ because nodes $i$ and $j$ may belong to different functional hierarchies. Thus, activity flow prediction is computed as follows:

$$E_j = \sum_{i \neq j \in V} A_i SC_{ji} \tag{9}$$

In addition, the combination of spatial and functional embeddings can be deduced as follows:

$$E_j = \sum_{i \neq j \in V} A_i SC_{ji} \frac{1}{D_{ij}} \tag{10}$$

### Statistics analysis

To evaluate the influence of activity flow prediction with different routes, a one-way repeated measures ANOVA was conducted. A post-hoc analysis was performed to assess the differences in prediction accuracy for each of the two routing protocols using paired sample $t$-tests. We set a statistically significant threshold of $p < 0.05$, with Bonferroni correction (i.e., $p < 0.05/n$), where $n$ indicates the number of statistical comparisons for a specific measure. Based on strategies commonly used in brain network studies [82,112], we performed statistical analysis for the AUC of prediction accuracy across all density thresholds, which provides a summarized measure independent of a single threshold selection. The statistical analysis was performed using SPSS 23.

### Supporting information

**S1 Fig. Accuracy of the activity flow prediction based on the direct (FC) and shortest (SPL$_{wei}$ and SPL$_{bin}$) paths with an extension of the network density to 100%.** Statistical comparisons were performed for the main (A), replication (B), and reduced (C) datasets. Statistical significance was identified based on the area under the curve (AUC) across all density thresholds. FC,

functional connectivity; $SPL_{wei}$, shortest path length based on weighted network; $SPL_{bin}$, shortest path length based on binary network. ***$p < 0.001$ ($p < 0.05$, Bonferroni corrected).
(TIF)

**S2 Fig. Accuracy of the activity flow prediction based on the direct path (FC) and shortest path ($SPL_{wei}$ and $SPL_{bin}$) for our working memory dataset.** Statistical significance was identified based on the area under curve (AUC) across all sparsity thresholds. FC, functional connectivity; $SPL_{wei}$, shortest path length based on weighted network; $SPL_{bin}$, shortest path length based on binary network; and ***$p < 0.001$ ($p < 0.05$, Bonferroni corrected).
(TIF)

**S3 Fig. Accuracy of the activity flow prediction based on the shortest path and other network communication strategies for our working memory dataset.** (A) For the weighted network, the accuracy of activity flow prediction based on the shortest path and other network communication strategies. (B) For the binary network, the accuracy of activity flow prediction based on the shortest path and other network communication strategies for all datasets. Statistical significance was identified based on the area under curve (AUC) across all sparsity thresholds. $X_{wei}$, routing metric X calculated based on the weighted network; $X_{bin}$, routing metric calculated based on the binary network; ***$p < 0.001$ ($p < 0.05$, Bonferroni corrected);
(TIF)

**S4 Fig. Effects of spatial and functional embedding of routes on activity flow prediction for our working memory dataset.** Statistical comparisons of the accuracy of activity flow prediction before and after considering the spatial embedding (A), functional embedding (B), and the both embeddings (C). Statistical significance was identified based on the area under curve (AUC) across all sparsity thresholds. FC, functional connectivity; SE, spatial embedding; FE, functional embedding; $SPL_{wei}$, shortest path length based on weighted network; $SPL_{bin}$, shortest path length based on binary network; and ***$p < 0.001$ ($p < 0.05$, Bonferroni corrected).
(TIF)

**S5 Fig. Performance gains in activity flow prediction after combining spatial and functional embedding of routes for our working memory dataset.** Accuracy of the activity flow prediction based on FC (A), $SPL_{wei}$ (B), and $SPL_{bin}$ (C) with different embeddings. Statistical significance was identified based on the area under curve (AUC) across all sparsity thresholds. FC, functional connectivity; FE, functional embedding; SE, spatial embedding; $SPL_{wei}$, shortest path length based on weighted network; $SPL_{bin}$, shortest path length based on binary network; ns, nonsignificant. ***$p < 0.001$ ($p < 0.05$, Bonferroni corrected).
(TIF)

**S6 Fig. Performance of the activity flow prediction with different routes that combined spatial and functional embedding for our working memory dataset.** Statistical significance was identified based on the area under curve (AUC) across all sparsity thresholds. FC, functional connectivity; FE, functional embedding; SE, spatial embedding; $SPL_{wei}$, shortest path length based on weighted network; $SPL_{bin}$, shortest path length based on binary network; ns, nonsignificant. *$p < 0.05$, ***$p < 0.001$ ($p < 0.05$, Bonferroni corrected).
(TIF)

**S7 Fig. Validation analysis with another parameter $k = 10$ of path ensembles for main datasets.** Statistical significance was identified based on the area under curve (AUC) across all sparsity thresholds. $X_{wei}$, routing metric X calculated based on the weighted network; $X_{bin}$, routing metric calculated based on the binary network; ns, nonsignificant. ***$p < 0.001$ ($p < 0.05$, Bonferroni corrected).
(TIF)

**S8 Fig. Validation analysis with another parameter $k = 10$ of path ensembles for our working memory dataset.** Statistical significance was identified based on the area under curve (AUC) across all sparsity thresholds. $X_{wei}$, routing metric X calculated based on the weighted network; $X_{bin}$, routing metric calculated based on the binary network. ***$p < 0.001$ ($p < 0.05$, Bonferroni corrected).
(TIF)

**S9 Fig. Effects of network density on activity flow prediction for different routing protocols.** To keep the same density with the original FC network, SPL networks were sparsified according to path weights directly (A-C), with direct paths and indirect paths preserved proportionally (D-F), and with only indirect paths preserved (G-I). Statistical significance was identified based on the area under the curve (AUC) across all density thresholds. FC, functional connectivity; $SPL_{wei}$(SD), shortest path length based on weighted network and having the same density with the original FC network; $SPL_{bin}$(SD), shortest path length based on binary network and having the same density with the original FC network. ***$p < 0.001$ ($p < 0.05$, Bonferroni corrected).
(TIF)

## Acknowledgments

Public dataset for main analysis was provided by the Human Connectome Project, Washington University–Minnesota Consortium (principal investigators David Van Essen and Kamil Ugurbil) funded by the 16 NIH institutes and centers that support the NIH Blueprint for Neuroscience Research and by the McDonnell Center for Systems Neuroscience at Washington University.

## Author contributions

**Conceptualization:** Zhengdong Wang, Dazhi Yin.

**Data curation:** Zhengdong Wang, Yifeixue Yang, Ziyi Huang, Wanyun Zhao, Kaiqiang Su, Hengcheng Zhu.

**Formal analysis:** Zhengdong Wang, Wanyun Zhao.

**Funding acquisition:** Dazhi Yin.

**Investigation:** Zhengdong Wang.

**Methodology:** Zhengdong Wang, Dazhi Yin.

**Software:** Zhengdong Wang.

**Validation:** Zhengdong Wang.

**Visualization:** Zhengdong Wang.

**Writing – original draft:** Zhengdong Wang, Dazhi Yin.

**Writing – review & editing:** Zhengdong Wang, Yifeixue Yang, Ziyi Huang, Wanyun Zhao, Kaiqiang Su, Hengcheng Zhu, Dazhi Yin.

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
