## [Decision Letter · Decision Letter 0]

9 Dec 2024

PCOMPBIOL-D-24-01627

Exploring the transmission of cognitive task information through optimal brain pathways

PLOS Computational Biology

Dear Dr. Yin,

Thank you for submitting your manuscript to PLOS Computational Biology. After careful consideration, we feel that it has merit but does not fully meet PLOS Computational Biology's publication criteria as it currently stands. Therefore, we invite you to submit a revised version of the manuscript that addresses the points raised during the review process.

Please submit your revised manuscript within 60 days Feb 08 2025 11:59PM. If you will need more time than this to complete your revisions, please reply to this message or contact the journal office at ploscompbiol@plos.org. Please include the following items when submitting your revised manuscript:

We look forward to receiving your revised manuscript.

Kind regards,

Linden Parkes

Guest Editor

PLOS Computational Biology

Andrea E. Martin

Section Editor

PLOS Computational Biology

Feilim Mac Gabhann

Editor-in-Chief

PLOS Computational Biology

Jason Papin

Editor-in-Chief

PLOS Computational Biology

**Journal Requirements:**

5) Please amend your detailed Financial Disclosure statement. This is published with the article. It must therefore be completed in full sentences and contain the exact wording you wish to be published. Please ensure that the funders and grant numbers match between the Financial Disclosure field and the Funding Information tab in your submission form. Note that the funders must be provided in the same order in both places as well. State what role the funders took in the study. If the funders had no role in your study, please state: "The funders had no role in study design, data collection and analysis, decision to publish, or preparation of the manuscript.".

**Reviewers' comments:**

Reviewer's Responses to Questions

**Comments to the Authors:**

Reviewer #1: The authors provide a comprehensive overview of the prediction accuracy of different network definitions on activity flow mapping. The main results appear to demonstrate that direct path (defined as Pearson’s FC) is more accurate than other, indirect methods such as shortest path length. They test their results over a range of sparsity values for the original FC matrix and further demonstrate that the incorporation of spatial distance and functional asymmetry improve prediction accuracy. The manuscript is well written and the results are presented clearly. I do have a few comments, below:

The reported methods on the activity flow paradigm are poorly reported. From the manuscript, it is not clear what Ai represents (e.g. beta-weights?) and there is no mention of the leave-one-region-out simulations that I assume must take place for the comparison between prediction and actual activity to be meaningful.

My major criticism of the framework presented deals with the sparsity of the compared networks. The authors define sparsity for their direct-path network (FC). From this sparse FC network, they then compute their other (in-direct) networks (e.g. SPLwei/SPLbin). Because the in-direct networks consider in-direct connections, the sparsity of the resulting SPL networks decreases (see Figure 2). It seems to me that across all sparsity levels tested, the SPLwei networks will be nearly fully connected in each case, and as the sparsity of the original FC network is increased, they will be generated from networks with higher signal-to-noise ratio. Thus it makes sense that the prediction accuracy of the SPLwei networks will continue to increase, even while the FC begins to plateau, as the SPL edges are less polluted by erroneous connections (see Figure 3). In fact, it appears possible that if the authors extended their analysis beyond sparsity levels of 0.25, SPLwei may overtake FC in prediction accuracy. I believe that the manuscript would benefit from (a) extension of sparsity beyond 0.25 to see if this is the case, (b) discussion of the above limitations, and (c) an analysis that actually keeps the sparsity of the resulting networks consistent by applying the same sparsity thresholds to the indirect networks.

Reviewer #2: In “Exploring the transmission of cognitive task information through optimal brain pathways”, Wang et al. build on the activity flow framework to investigate the graph routing protocols underlying the transmission of cognitive information in the human brain. The study is highly original and builds bridges between typically separate areas of neuroscience, such as graph theory and cognitive task brain activations. The general approach is highly promising, yet there are some flaws with the conceptualization of the activity flow modeling procedures. See below for details, but to summarize, the main issue is that routing protocols are allowed to modify connectivity estimates, violating basic physical principles for how activity flows propagate in actual brains. For example, shortest paths are calculated based on standard walks over FC graphs (much as flows actually occur in the brain), but then the estimated shortest path lengths are used to create new connections weighted by those paths. It is unclear whether the authors think these new connections actually exist, or whether they are representing something more abstract than actual new connections. In any case, the results are difficult to comprehend from a simple activity propagation perspective, due to graph measures on a given graph actually modifying that graph for subsequent activity propagation calculations. Instead, I suggest the authors consider keeping the graph stable and simulate flows according to the given communication protocol (e.g., shortest path) being tested.

Major concerns:

• It is unclear what “sparsity” means in the plots. Are there fewer connections with more “sparsity” in the plots, or fewer? Often density is used (rather than sparsity) for thresholding, so I want to make sure sparsity was really meant (rather than density). I assume “density” is meant whenever the term “sparsity” was used. For instance, 0.15 is used as an example sparsity in the text, which would correspond (if 0.15 is to be converted to a percentage) to a density of 85% (1-0.15=0.85*100). That is quite dense. Also, it seems that the density (or sparsity) values should be reported in terms of percentages (rather than decimal point values).

• The biggest issue I see with this study is that the main analysis (Figure 1) assumes a difference between a “direct” flow route and a “shortest path” flow route, while those are often the same thing. The authors appear to force a difference between “direct” and “shortest path”, but this just makes their “shortest path” no longer the true shortest path. Ultimately, this sets up a straw man argument, wherein the “shortest path” communication protocol is destined to fail. Why exclude the true shortest path (the direct path) when considering the shortest path?

• However, careful consideration of Figure 2b (and the Methods) suggests the “shortest path” may be something more complex, with the direct paths and multi-step paths all contributing simultaneously. This is confusing, as Figure 1 illustrated a more circumscribed definition of “direct” and “shortest path”. Instead of having all regions’ activity making a weighted contribution to the target, it seems that activity flows should be simulated as actual flow processes, rather than simply modifying the FC/weight values. While clever, only modifying the FC values (rather than simulating alternate flow routes) confuses direct and alternate-route processes. Perhaps most problematically, all sources are included (using a leave-one-out approach), making it such that the direct path should always be the best routing protocol. This is because only direct paths actually (in reality, outside the models) have access to the target, with all other signals needing to pass through the direct sources. What is needed is likely a different activity flow routing setup, such as is used in several recent studies (see Ito T, Yang GR, Laurent P, Schultz DH, Cole MW (2022) Constructing neural network models from brain data reveals representational transformations linked to adaptive behavior. Nat Commun 13:673. And Cocuzza CV, Sanchez-Romero R, Ito T, Mill RD, Keane BP, Cole MW (2024) Distributed network flows generate localized category selectivity in human visual cortex Kay K, ed. PLoS Comput Biol 20:e1012507.). These studies start with a subset of activations, then simulate activity flows over multiple steps to test the efficacy of multi-step activity flow processes. In contrast, the current study’s approach simulates flow processes as if they were all direct, simply adding a connection from a source to a target with a weight reflecting the length of a shortest path from that source to that target. This violates the organization of the estimated connectivity graphs, which requires that indirect signals must pass through direct sources on the way to a given target. That said, there may be some clever way to think about the problem that does not require simulations that are physically realistic (in the sense of only direct sources directly impacting targets), but if so this should be made much clearer in the manuscript.

• The results in Figure 3 suggest that the results are highly dependent on network density thresholds. In this case at a 0.24 density the direct and shortest path results converged. In general, many results are dependent on network density, raising concerns about the robustness of the results.

Minor concerns:

• It is stated that “the shortest path routing is unrealistic for decentralized nervous systems because it requires individual elements to have knowledge of the global network topology”. However, Misic et al. (2015) (Mišić B, Betzel RF, Nematzadeh A, Goñi J, Griffa A, Hagmann P, Flammini A, Ahn Y-Y, Sporns O. 2015. “Cooperative and Competitive Spreading Dynamics on the Human Connectome”. Neuron. 86:1518–1529.) showed that shortest paths are special even in a decentralized system because signals arrive first via shortest paths even with simple diffusion processes. Thus, shortest paths are not properly characterized here.

• All negative connections were set to 0 because of “the lack of explicit meaning”. It would be important to make it cleaer what is meant here, since negative connections do indeed have a meaning in theory (inhibitory connectivity).

• Use of Pearson correlation is a problem, as a “direct” route has a high probability of not existing. See Reid AT, Headley DB, Mill RD, Sanchez-Romero R, Uddin LQ, Marinazzo D, Lurie DJ, Valdés-Sosa PA, Hanson SJ, Biswal BB, Calhoun V, Poldrack RA, Cole MW. 2019. “Advancing functional connectivity research from association to causation”. Nat Neurosci. PMID: 31611705. It would be important to acknowledge limitations of using Pearson correlation as a functional connectivity measure.

• It would be important to see the accuracies of the activity flow predictions in the text, so the reader can better interpret the results. It would also be helpful to see the accuracies in terms of r-values, rather than AUC, since r-values are easier to understand.

**Have the authors made all data and (if applicable) computational code underlying the findings in their manuscript fully available?**

Reviewer #1: Yes

Reviewer #2: Yes

PLOS authors have the option to publish the peer review history of their article (what does this mean? ). If published, this will include your full peer review and any attached files.

**Do you want your identity to be public for this peer review?** For information about this choice, including consent withdrawal, please see our Privacy Policy .

Reviewer #1: **Yes: ** S. Parker Singleton

Reviewer #2: No

**Figure resubmission:**
---

## [Decision Letter · Decision Letter 1]

12 Feb 2025

Dear Dr. Yin,

We are pleased to inform you that your manuscript 'Exploring the transmission of cognitive task information through optimal brain pathways' has been provisionally accepted for publication in PLOS Computational Biology.

Best regards,

Linden Parkes

Guest Editor

PLOS Computational Biology

Andrea E. Martin

Section Editor

PLOS Computational Biology

Reviewer's Responses to Questions

**Comments to the Authors:**

Reviewer #1: I would like to thank the authors for their thorough response to my comments. My concerns have been addressed.

Reviewer #2: All of my main concerns have been addressed.

**Have the authors made all data and (if applicable) computational code underlying the findings in their manuscript fully available?**

Reviewer #1: Yes

Reviewer #2: Yes

PLOS authors have the option to publish the peer review history of their article (what does this mean? ). If published, this will include your full peer review and any attached files.

**Do you want your identity to be public for this peer review?** For information about this choice, including consent withdrawal, please see our Privacy Policy .

Reviewer #1: **Yes: ** S. Parker Singleton

Reviewer #2: No

---

## [Editor Report · Acceptance letter]

PCOMPBIOL-D-24-01627R1

Exploring the transmission of cognitive task information through optimal brain pathways

Dear Dr Yin,

I am pleased to inform you that your manuscript has been formally accepted for publication in PLOS Computational Biology. Your manuscript is now with our production department and you will be notified of the publication date in due course.

With kind regards,

Zsofia Freund
